# Advanced Nanopharmaceutical Intervention for the Reduction of Inflammatory Responses and the Enhancement of Behavioral Outcomes in APP/PS1 Transgenic Mouse Models

**DOI:** 10.3390/pharmaceutics17020177

**Published:** 2025-01-31

**Authors:** Jun Li, Dongqing Huang, Wanchen Liao, Yulin Wang, Yibiao Liu, Ping Luan

**Affiliations:** 1Department of Alzheimer’s Disease Clinical Research Center, The Affiliated Guangdong Second Provincial General Hospital of Jinan University, Guangzhou 510317, China; 2100243044@email.szu.edu.cn (J.L.); 2100243052@email.szu.edu.cn (D.H.); 2200243071@email.szu.edu.cn (W.L.); 2300243042@email.szu.edu.cn (Y.W.); 2School of Basic Medical Sciences, Health Science Center, Shenzhen University, Shenzhen 518060, China; 3The Second School of Clinical Medicine, Southern Medical University, Guangzhou 510260, China; 4Longgang Central Hospital of Shenzhen, Shenzhen 518116, China

**Keywords:** Alzheimer’s disease, curcumin, black phosphorus, inhibition of Aβ aggregation, 4-(Dimethylamino) cinnamic acid

## Abstract

**Background**: The excessive accumulation of Aβ plays a critical role in the development of Alzheimer’s disease. However, the therapeutic potential of drugs like curcumin is often limited by low biocompatibility and BBB permeability. In this study, we developed a nanomaterial, BP-PEG-Tar@Cur, which was designed to enhance the biocompatibility of (curcumin) Cur, target Aβ, and augment BBB permeability through near-infrared (NIR) photothermal effects. **Methods**: Soluble Aβ, ThT fluorescence, and Aβ depolymerization fluorescence experiments were conducted to evaluate the ability of BP-PEG-Tar@Cur to inhibit Aβ aggregation and dissociate Aβ fibrils. Cell uptake assays were performed to confirm the targeting ability of BP-PEG-Tar@Cur towards Aβ. In vitro mitochondrial ROS clearance and in vivo detection of inflammatory factors were used to assess the anti-inflammatory and antioxidant properties of the nanodrug. Water maze behavioral experiments were conducted to evaluate the effect of BP-PEG-Tar@Cur on spatial memory, learning ability, and behavioral disorders in AD mice. **Results**: The nanodrug effectively inhibited Aβ aggregation and dissociated Aβ fibrils in vitro. BP-PEG-Tar@Cur demonstrated efficiency in curbing ROS overproduction in mitochondria and dampening the activation of microglia and astrocytes triggered by Aβ aggregation. Water maze behavioral experiments revealed that BP-PEG-Tar@Cur enhanced spatial memory, learning ability, and alleviated behavioral disorders in AD mice. **Conclusions**: Collectively, these findings demonstrate that BP-PEG-Tar@Cur has the potential to be an effective targeted drug for inhibiting Aβ aggregation and improving cognitive impairment in AD mice.

## 1. Introduction

Alzheimer’s disease (AD) is the predominant form of dementia, constituting approximately 60% to 80% of all cases. From 2000 to 2019, there has been a significant rise in AD-related deaths, with a notable surge during the Coronavirus Disease 2019 (COVID-19) pandemic [1]. As the population ages, particularly with the baby boom generation reaching over 65, AD is emerging as one of the most burdensome and fatal diseases [2]. Moreover, AD has become the third leading cause of death and disability among the elderly, surpassed only by cardiovascular and cerebrovascular diseases, as well as malignant tumors [3]. AD is a multi-factorial disease, implicating aspects such as amyloid beta (Aβ) peptides aggregation, tau protein hyperphosphorylation, inflammatory responses, reactive oxygen species accumulation and cholinergic dysfunction, etc. [4,5]. Among these etiological mechanisms, the amyloid cascade hypothesis is most widely accepted [6]. The hypothesis underscores the abnormal accumulation and folding of amyloid beta outside the neurons in the brain as a critical pathological feature of AD [7]. The excessive production and misfolding of amyloid beta peptides play a pivotal role in the formation of amyloid plaques [8]. These plaques are considered to trigger a series of subsequent adverse reactions, including inflammatory responses, progressive synaptic injury and tau hyperphosphorylation, ultimately leading to widespread neuronal dysfunction and death [9]. Given the significance of Aβ fibrils and plaques in inducing pathological alterations and cognitive deficits, drugs candidates capable of inhibiting aggregation of Aβ present hopeful prospects for effective AD treatment.

Curcumin (Cur), a natural polyphenolic compound, has garnered significant interest among researchers due to its profound anti-inflammatory, antioxidant, inhibitory properties against Aβ folding and fibril formation [10,11]. However, its therapeutic potential is hindered by its poor bioavailability and limited aqueous solubility [12]. Another obstacle is the blood–brain barrier (BBB), a protective mechanism that restricts the entry of most substances, resulting in extremely low drug permeability [13,14]. To address these challenges, numerous research strategies have been explored. Polyethylene glycol (PEG), a hydrophilic polymer, has been extensively utilized in nanodrug delivery systems due to its low immunogenicity, biocompatibility, and strong clinical conversion ability [15,16,17]. Black phosphorus (BP) nanosheets has a superior loading ability and absorption efficiency compared to other 2D nanomaterials, owing to its higher surface-to-volume ratio [18]. Notably, a considerable number of studies have demonstrated that BP nanosheets enabled increased drug permeability through opening the BBB photothermally, leveraging their excellent photothermal properties under near-infrared (NIR) irradiation [19]. Most importantly, BP also has the unique ability to suppress Aβ self-aggregation under photothermal conditions and can decompose into physiologically compatible phosphates and phosphites [20]. Compared to other penetrating and targeting transport peptides, BP exhibits greater stability, reducing immune system clearance and thus optimizing therapeutic effectiveness [18]. Recent studies have identified a series of cinnamic acid derivatives with effective anti-cancer, anti-inflammation and antioxidant properties, as well as the ability to improve the pathological condition of AD models [21,22,23]. Some studies proposed that the probe containing the dimethyl amino group has higher affinity for the Aβ aggregates [24], suggesting that cinnamic acid derivatives may offer a concurrent approach to achieve anti-inflammatory, antioxidant ability and enhanced aiming function.

In this work, a nanodrug with a potential treatment effect was successfully synthesized by integrating the BP nanosheets, curcumin, and 4-(Dimethylamino) cinnamic acid. The synthesis approach for BP-PEG-Tar@Cur entails harnessing the photothermal properties of black phosphorus (BP) to permeabilize the blood–brain barrier (BBB), modifying curcumin (Cur) with polyethylene glycol (PEG) for enhanced biocompatibility, and integrating 4-(Dimethylamino) cinnamic acid (Tar), a derivative of cinnamic acid, to bolster the targeting precision towards amyloid-beta (Aβ) plaques. The partial therapeutic effects of this nanodrug have been concisely and clearly illustrated in the graphical abstract. Our finding revealed that BP-PEG-Tar@Cur effectively suppressed the excessive inflammatory response triggered by Aβ aggregation in astrocytes and microglia. In addition, this nanodrug not only inhibited Aβ aggregation, but also facilitated the disassembly of existing Aβ aggregates. Most importantly, the inclusion of 4-(Dimethylamino) cinnamic acid and a BP nanosheet significantly enhanced the targeting of Aβ and BBB penetrating rate. The beneficial impact of BP-PEG-Tar@Cur on spatial learning and memory impairment was verified in an APP/PS1 mice model using a behavioral test (Morris water maze). This nanodrug, with the capability of photothermal BBB penetration and Aβ targeting, can actively ameliorate the Aβ burden, representing a promising Aβ aggregation inhibitor for AD treatment.

## 2. Materials and Methods

### 2.1. Materials

Curcumin (99.68%), 4-(Dimethylamino) cinnamic acid (99.99%), N-Hydroxy succinimide (NHS) (98%) were purchased from Bide Pharmatech Co., Ltd. (Shanghai, China). DSPE-PEG2000-NH_2_ was obtained from Weihua Biotechnology Co., Ltd. (Guangzhou, China). The black phosphorus nanosheet was supplied by Shenzhen Zhongke Mo Phosphorus Technology Co., Ltd. (Shenzhen, China). The lyophilized form of Aβ1−42 peptide (>95%) was purchased from Shanghai Yuanye Bio-Technology Co., Ltd. (Shanghai, China). Hexafluoroisopropanol (HFIP) (99.8%) was acquired from Shanghai Meryer Chemical Technology Co., Ltd. (Shanghai, China). DAPI was purchased from Beijing Labgic Technology Co., Ltd. (Beijing, China). MitoSOX red mitochondrial superoxide indicator was obtained from Thermo Fisher, Invitrogen (Carlsbad, CA, USA). Calcein-AM/PI Double Stain Kit was purchased from Yeasen Biotechnology Co., Ltd. (Shanghai, China). 1-(3-Dimethylamino propyl)-3-ethylcarbodiimide hydrochloride (EDC), N, N-Dimethylformamide (DMF) (99.8%) and Thioflavin T (ThT) were provided by Shanghai Aladdin Biochemical Technology Co., Ltd. (Shanghai, China). The Cell Counting Kit-8 was purchased form Abbkine Scientific Co., Ltd. (Wuhan, China). BeyoECL Plus (Ultrasensitive ECL kit) was obtained from Beyotime Biotechnology (Shanghai, China). Antibodies against amyloid beta (Aβ), amyloid precursor protein (APP), Presenilin-1 (PS1), glial fibrillary acidic protein (GFAP), ionized calcium binding adapter protein 1 (Iba-1), Glyceraldehyde-3-phosphate dehydrogenase (GAPDH), Alexa Fluor 647 (AF647) were purchased from Cell Signaling Technology (Danvers, MA, USA). The antibodies against the beta-site APP-cleaving enzyme 1 (BACE1) and the horseradish peroxidase (HRP)-conjugated secondary antibody were purchased from Vicbio Biotechnology Co., Ltd. (Beijing, China). All other commonly used reagent and consumables were obtained from Shanghai Epizyme Biomedical Technology Co., Ltd. (Shanghai, China). and Thermo Fisher, Gibco (Carlsbad, CA, USA).

### 2.2. Preparation of the Nanoparticles

The synthesis of PEG-Tar was achieved through the employment of the EDC/NHS reaction. A molar ratio of 1:1:1 was used, combining 38 mg of 4-(Dimethylamino) cinnamic acid (Tar), 23 mg of N-Hydroxy succinimide (NHS), and 38 mg of 1-(3-Dimethylaminopropyl)-3-ethylcarbodiimide hydrochloride (EDC) in a round-bottom flask. Subsequently, 2 mL of DMF was added, and the mixture was stirred using a magnetic stirrer to activate the (carboxylic acid group) COOH of Tar via the EDC/NHS reaction. After 5 h, 200 mg of DSPE-PEG2000-NH_2_ (PEG) was introduced, enabling the EDC/NHS reaction to facilitate the connection of the COOH bond on the activated Tar with the NH_2_ bond on the PEG, thereby resulting in the synthesis PEG-Tar. The subsequent phase of the synthesis involved the formation of PEG-Tar@Cur through a hydrophobic reaction. A total of 10 mg of Cur was added to the PEG-Tar solution and stirred for 12 h. Subsequently, the entire reaction solution was gradually added to 12 mL of water while rapidly stirring and heating at 40 °C until all DMF had evaporated. During this process, the Cur was encapsulated within the hydrophobic ball formed by the C_18_ chain on PEG. Subsequently, the insoluble Cur was separated from the solution by centrifugation to purify the PEG-Tar@Cur. This was accomplished by transferring the solution to a 15 mL centrifuge tube and collecting the supernatant contained PEG-Tar@Cur after centrifuging at 6000 rpm for 5 min. The final synthesis of BP-PEG-Tar@Cur utilizing phosphorus–oxygen bonding entailed the resuspension of 4 mL BP (NMP,1-Methyl-2-pyrrolidinone) (1 mg/mL) in ultrapure water after centrifugation and added to the PEG-Tar@Cur solution. After 12 h, the reaction solution was extracted and centrifuged at 7000 rpm for 5 min to remove the uncombined BP. The supernatant was removed after centrifugation, and 5 mL of ultrapure water was added to resuspend the remaining material. This step was repeated to ensure the thorough washing of any unbound PEG-Tar@Cur. Ultimately, the BP-PEG-Tar@Cur suspension was obtained by resuspension in ultrapure water.

### 2.3. Characterization of the Nanoparticles

The zeta potentials and size distribution of the synthetic particles were determined by Malvern Zetasizer Pro (Malvern, UK). The morphology was analyzed with a FEI Talos F200 X G2 transmission electron microscope (TEM) in Thermo Fisher (Waltham, MA, USA). A small amount of the sample was dispersed in ethanol, and then a few drops were added to the copper net. After drying, the sample was imaged under the condition of an accelerating voltage of 200 kV. Fourier transform infrared spectroscopy (FT-IR) analysis was conducted on the Science Nicolet iS20 in Thermo Fisher (Waltham, MA, USA). Prior to analysis, the attenuated total reflection (ATR) accessory in the spectrometer’s optical path performed an initial scan of the air background in a moisture-free environment. Subsequently, a sample droplet was placed on the ATR accessory’s crystal face, enabling the collection of the infrared spectrum. The instrument was calibrated to execute 32 scans with a resolution of 4 cm^−1^ over a wavenumber range of 400–4000 cm^−1^. Ultraviolet spectrum was observed under LabSolutions UV-Vis (UV-26001) (SHIMADZU, Kyoto, Japan). Raman spectra was determined using an alpha300 R (WITec, Ulm, Germany) with a 532 nm laser, an Olympus 20× objective with a laser energy of 36 mW, 1800 g/mm grating.

### 2.4. Aβ Monomer and Aβ Fibrils Preparation

Aβ1-42 monomers were prepared by treating the Aβ1-42 peptide with pure HFIP. HFIP acts as a β-fold-breaking solvent that enhances the amyloid proteinogenicity of Aβ1-42, resulting in a more homogeneous and monomerized Aβ1-42 [25]. Prior to use, cryopreserved Aβ1-42 was brought to room temperature for 30 min to avoid peptide condensation. Then, 222 μL HFIP was added to obtain 1 mM solution of Aβ1-42. This solution was sonicated in an ice water bath for 10 min and then incubated at 4 °C with shaking for 2 h. After vortexing, the solution was aliquoted into 10 μL fractions. All aliquots were placed in a SpeedVac at 45 °C for 30 min to eliminate any residual Hexafluoroisopropanol (HFIP). Finally, the resulting Aβ1-42 monomers preparations were stored in EP tubes at −20 °C until required. For utilization, the aliquots were thawed at room temperature for 10 min and then mixed with 10 μL of dimethyl sulfoxide (DMSO) in an ultrasonic ice water bath for another 10 min. Subsequently, the mixture was diluted to the desired concentration with phosphate-buffered saline (PBS) (pH 7.4, 1×) [26]. The Aβ1-42 monomer solution was then gently agitated and incubated at 37 °C for 72 h to facilitate the formation of fibrils [27].

### 2.5. Drug Loading Capacity

Cur (1 mg) was dissolved in absolute ethanol to prepare different diluted concentrations (0.625, 1.2, 2.5, 5, 10, 20 μg/mL) [28]. The standard curve of Cur was obtained by measuring the absorbance value at 425 nm with an ultraviolet spectrophotometer. To evaluate the loading capacity of Cur, 20 μL of BP-PEG-Tar@Cur was diluted to 2 mL using absolute ethanol and its absorbance at 425 nm was measured. The encapsulation and drug loading capacities were calculated by substituting the standard curve into the appropriate equations.

### 2.6. In Vitro Release Profile

To assess the in vitro release profile of Cur, we followed a preciously described method [29]. PEG-Tar@Cur (0.5 mL) and BP-PEG-Tar@Cur (0.5 mL) was transferred to dialysis bags (MW cutoff: 1500 Da) and then immersed in 50 mL of absolute ethanol release medium. The setup was maintained at 37 °C with the use of a magnetic stirrer. At specific intervals (2, 4, 6, 8, 10, 12, 24 h), 3 mL of release solution was extracted and replaced with an equal volume of absolute ethanol. Subsequently, the absorbance value at 425 nm of the release solution was measured in order to calculate the cumulative release of Cur.

### 2.7. Photothermal Effect Research

To investigate the temperature change in BP-PEG-Tar@Cur under NIR irradiation in vitro, different concentrations of BP-PEG-Tar@Cur (0, 0.5, 2, 5, 10, 20 μg/mL) in PBS were irradiated by NIR (808 nm, 1 W/cm^2^) for 10 min. Then, thermal images were captured using a thermal imaging camera [30]. The in vivo photothermal effect research was performed in accordance with the methods mentioned below. The wild-type mice were divided into three groups: a control group, an NIR group and a BP-PEG-Tar@Cur+NIR group. Before the experiment began, the fur from the superior orbital margin to the posterior occipital lobe of the mice was shaved and cleaned with a hair removal cream to expose the clean scalp. The BP-PEG-Tar@Cur+NIR group was administered a combination of BP-PEG-Tar@Cur (Cur:1 mg/kg) and Evans blue (EB) solution (20 mg/mL) via injection. After one hour, the mice were anesthetized and positioned beneath the NIR laser probe for NIR irradiation (808 nm, 1 W/cm^2^) for 10 min. The control group and the NIR group underwent identical preparation procedures and received the same volume of physiological saline. However, it should be noted that only the NIR group received NIR irradiation, while the control group did not. Finally, the mice were euthanized, and their brains were dissected to observe the distribution of the Evans blue solution within the brain 24 h later.

### 2.8. Measurement of Soluble Aβ

The experimental procedures for soluble Aβ assays were adapted from a previous protocol [31]. Aβ1-42 (25 μM) was incubated with PBS, Cur, PEG-Tar@Cur and BP-PEG-Tar@Cur (Cur concentration of 0.5 μg/mL). Aβ monomers were used as a control group. After incubating for 72 h at 37 °C with shaking at 100 rpm, the samples were centrifuged at 20,000 rpm for 20 min. Finally, the concentration of soluble Aβ was determined through bicinchoninic acid assay (BCA) protein quantification.

### 2.9. ThT Fluorescence Determination

Aβ1-42 (25μM) was combined with 480 μL PBS, Cur, PEG-Tar@Cur, and BP-PEG-Tar@Cur (Cur concentration of 0.5 μg/mL), respectively, and then placed in a shaker at 37 °C, 100 rpm. From this mixture, 50 μL samples were extracted at predetermined time points (0, 2, 4, 6, 8, 10, 12, 24 h) and mixed with 450 μL ThT buffer (25μM) for subsequent analysis. The ThT fluorescence intensity was determined by a fluorescence spectrometer with excitation and emission wavelengths set at 440 nm and 485 nm, respectively, and a slit width of 5 nm at 25 °C [32].

### 2.10. Aβ Fibrils Depolymerization Fluorescence Imaging

Aβ1-42 samples (25 μM) were incubated for 24 h both in the absence and presence of the relevant drugs. Subsequently, 10 μL aliquots were deposited onto slides and allowed to air dry. The samples were then stained with ThT buffer (10 μM) for 10 min, rinsed gently with water, and air-dried as per established procedures [33]. Finally, imaging was performed with fluorescence microscopy (Olympus, Tokyo, Japan).

### 2.11. Cell Viability Assay

The viability of the cells exposed to nanomaterials was accessed using the CCK8 kit protocol. Briefly, a suspension of 100 μL N2 a and bEnd.3 cell containing approximately 5 × 10^4^ cells were seeded onto the 96-well plates. After incubation for 24 h at 37 °C, the medium was replaced. A new medium containing various concentrations of Cur, PEG-Tar@Cur, and BP-PEG-Tar@Cur was incubated for 48 h and 72 h, respectively. Thereafter, the medium was replaced again, and 10 μL of CCK8 solution was added to each well. Untreated cells served as a positive control group, while the blank group consisted solely of the fresh medium. The absorbance value at 450 nm was detected using a microplate reader.

### 2.12. In Vitro Aβ-Targeted Assay

A small quantity of medium was dispensed at the location where the climbing tablet was positioned, ensuring that the tablet adhered to the 24-well plate tightly due to surface tension. A total of 2.5 × 10^5^ N2 a cells/mL were seeded into the wells and then placed in the incubator to allow for cell adhesion. Once the cells had adhered, the following substances were introduced and incubated for 24 h: Aβ1-42 (25 μM), Cur, BP-PEG@Cur, PEG-Tar@Cur, and BP-PEG-Tar@Cur (0.5 μg/mL). Following incubation, the cells were fixed by 4% paraformaldehyde for 15 min. Then, the cells were penetrated with 0.5% Triton X-100 for 20 min at room temperature. A 30 min incubation at 37 °C was conducted using 10% goat serum for the purpose of blocking. Subsequently, the primary antibody was incubated overnight at 4 °C for 1 h, followed by the secondary antibody incubation at room temperature overnight. Finally, the cells were incubated with 4′,6-Diamidino-2′-phenylindole (DAPI) (10 μg/mL) in the dark for 4 min. Following three washes with PBS, the samples were sealed with an anti-fluorescence quencher and then examined under ultra-high resolution confocal microscopy (LSM880, ZEISS, Jena, Germany). In the images, Aβ was rendered in red (647 nm channel), Cur in green (FITC channel), and DAPI in blue (488 nm channel).

### 2.13. Mitochondrial ROS Elimination Assay

N2 a cells (2.5 × 10^5^/well) were seeded onto climbing tablets in a 24-well plate. Aβ monomers (25 μM), Cur and BP-PEG-Tar@Cur (0.5 μg/mL) were co-cultured for 24 h after cells adhesion. At the end of incubation, the cells were fixed using 4% paraformaldehyde for 15 min. The MitoSOX red mitochondrial superoxide indicator (MitoSOX) solution (5 mM) was equilibrated to room temperature and then diluted to a working concentration of 5 μM with PBS. The cells on the climbing tablets were covered with 1–2 mL of the probe working solution and incubated at 37 °C away from light for 10 min [34]. After gently washed thrice with pre-warmed PBS, the cells were treated with DAPI (10 μg/mL) in the darkness for 5 min. Fluorescence images were observed by an ultra-high resolution confocal microscopy (LSM880, ZEISS, Germany).

### 2.14. In Vitro BBB Model

In a configuration based on a prior experiment, bEnd.3 cells were cultured in a transwell apparatus (6-well insert device, 0.4 μm polyester membrane, 4.67 cm^2^ of cell growth area) to simulate the BBB with tight junctions [35]. 5 × 10^5^ bEnd.3 cells were added to the upper chamber of the transwell with 1.5 mL of medium and 2.6 mL in the lower chamber. The resistance value was ascertained using the MERS00002 (Millicell-ERS) cell resistance meter. Once the TEER (transendothelial electrical resistance) value reached 200 Ω·cm^2^, the respective materials were introduced into the upper chamber. In the BP-PEG-Tar@Cur + NIR group, irradiation was conducted for 10 min following the addition of the materials. Then, 2 h later, a portion of the lower chamber medium was sampled to detect the concentration of Cur. The status of cells after NIR irradiation (808 nm, 1 W/cm^2^) was analyzed using Clacein-AM/PI staining, and the images were observed with fluorescence microscopy (Olympus, Japan).

### 2.15. Cell Culture and Animal Drug Administration

Mouse-derived neuroblastoma cells (N2 a) were cultivated in a mixture of 47% Dulbecco’s modified Eagle’s medium (DMEM), 47% opti-modified Eagle’s medium (Opti-MEM), 5% fetal bovine serum (FBS), and 1% penicillin–streptomycin. Mouse brain microvascular endothelial cells (bEnd.3) were cultured in 89% DMEM, 10% FBS, and 1% penicillin–streptomycin. All cell cultures were maintained in a humidified incubator at 37 °C with 5% CO_2_.

APP/PS1 mice were purchased from Zhuhai BesTest Bio-Tech Co., Ltd. (Zhuhai, China) and were maintained at a controlled temperature (22 ± 2 °C) under a 12 h light–dark cycle, with unlimited access to food and water. The wild-type mice served as the control group. The mice were randomly divided into four groups: control, AD, Cur, and BP-PEG-Tar@Cur+NIR. The mice in the Cur group and the BP-PEG-Tar@Cur+NIR group received a single tail vein injection of 1 mg/kg, while the control group and AD group were injected with an equivalent volume of saline. Subsequently, the BP-PEG-Tar@Cur+NIR group underwent the previously described surgical preparation and was subjected to NIR irradiation 10 min post-administration. All experimental protocols were conducted in strict accordance with the management regulations of the Institutional Animal Care and Use Committee of Shenzhen University Medical School (IACUC).

### 2.16. Morris Water Maze (MWM)

The Morris water maze (MWM) experiment represents a well-established methodology for evaluating the memory and learning ability of mice. The water maze analysis system consists of a cylindrical pool (120 cm in diameter, 50 cm in height) and a Morris water maze video analysis system.

Before the experiment, water was added to the pool until the water level was approximately 0.8 cm above the platform. Titanium dioxide powder was then added to the water and stirred evenly to whiten the water surface, making the black mice the focal point. The platform had a diameter of 5 cm and its surface was painted white in order to achieve the purpose of hiding the platform. The pool was equally divided into four quadrants, east, south, west and north, each with distinct markers on the wall. To facilitate acclimation, the mice were placed into the pool (without a platform) and permitted to swim freely for 2 min. The first part was the position navigation experiment, which mainly reflects the ability of memory and objects recognition of the mice. This experiment lasted for a total of 5 days, with four training sections conducted daily at a fixed time. Initially, the platform was situated in the NW quadrant, and then mice were introduced from various starting position facing the wall. The analytical system recorded the time taken to locate the platform (escape latency) and their swimming routes. Regardless of whether the mice found the platform within 60 s, they were allowed a 10 s rest before the next training. The average latency from the four daily training sessions was used as a metric for learning performance on that day. The second part is a detection test to evaluate the spatial memory ability of the mice. On the sixth day, the platform was removed and the mice were placed in the SE quadrant. The number of times the mice crossed the original platform within 1 min observation period were recorded.

### 2.17. Tissue Preparation

After the behavioral experiments, the mice were anesthetized with 0.3% pentobarbital sodium and then perfused with pre-chilled saline. For Western blotting (WB), the hippocampus was stored at −80 °C, separately, for later use. For immunofluorescence (IF) staining and hematoxylin and eosin (HE) staining, the entire brain was excised and fixed in 4% paraformaldehyde for 24 h. The brains were then dehydrated in 30% sucrose for 48 h and sectioned into 8-μm slices.

### 2.18. Western Blotting

The hippocampal samples were weighed and chilled on ice. Strong radioimmunoprecipitation assay (RIPA) lysate containing phenylmethanesulfonyl fluoride (PMSF) and protease inhibitors were added in the samples in a 1:10 mass to volume ratio. The samples were then homogenized electronically, followed by thorough lysis with an ultrasonic cell disruptor. After centrifugation at 14,000 rpm at 4 °C for 10 min, the supernatant was reserved for protein quantification. Before loading, the samples were subjected to boiling to denature proteins. Electrophoresis was performed using a 15% separation gel and then transferred proteins to polyvinylidene difluoride (PVDF) membranes afterwards. Desired protein bands were isolated from the PVDF membrane and blocked with protein-free rapid blocking solution. The primary antibodies (APP 1:1000; BACE1 1:2000; PS1 1:1000; Aβ 1:1000; GAPDH 1:1000) were added and incubated overnight at 4 °C. On the subsequent day, after retrieving the primary antibodies, fluorescent secondary antibodies were added and incubated at room temperature for 90 min. For the development of the blot, an ultrasensitive enhanced chemiluminescence (ECL) chemiluminescence kit was utilized to detect and amplify the fluorescent signals. The blot images were then processed and analyzed for grayscale value using Chemstudio (Analytik Jena, Jena, Germany), providing quantitative data on protein expression levels.

### 2.19. Hematoxylin and Eosin (HE) Staining

After completing the behavioral experiment, the brain, heart, liver, spleen, lung and kidney were harvested for histological analysis. These organs were then fixed in 4% paraformaldehyde and then sectioned into 6-μm thick paraffin slices. Following deparaffinization and rehydration, the slices underwent hematoxylin and eosin staining. Histological alterations were observed and images were captured by an Aperio CS2 scanner (Wetzlar, Germany, Leica, Germany).

### 2.20. Immunofluorescence Staining

The frozen slices were allowed to thaw to room temperature. The membrane was penetrated with 0.5% Triton X-100 at room temperature for 1 h, followed by a 30 min incubation with 10% goat serum at 37 °C for blocking. Slices were then incubated with primary antibodies (GFAP 1:200; Iba-1 1:100) overnight at 4 °C. After primary antibody retrieval on the following day, the sections were incubated with fluorescent secondary antibody for 1 h at room temperature. Following DAPI staining, fluorescence images were captured via ultra-high resolution confocal microscopy LSM880 from ZEISS (Oberkochen, Germany).

### 2.21. Statistical Analysis

All data were represented as means ± standard error of the medium (SEM). The statistical difference among multiple groups were assessed by one-way analysis of variance (ANOVA), followed by Dunnett’s multiple comparisons test. Two-group comparisons were conducted using unpaired *t*-tests. A *p*-value less than 0.05 was considered statistically significant. All data analyses and graphs were generated using GraphPad Prism 9 (GraphPad Software, San Diego, CA, USA).

## 3. Results and Discussion

### 3.1. Preparation and Characterization of the Nanodrug

The synthesis route of BP-PEG-Tar@Cur is illustrated in the graphical abstract. In brief, PEG-Tar@Cur was synthesized through activation of the carboxyl group on the target 4-(Dimethylamino) cinnamic acid and the amino group on PEG through EDC/NHS reaction. Subsequently, Cur was encapsulated with PEG-Tar using the hydrophobic interaction facilitated by the C_18_ chain on PEG, resulting in the formation of PEG-Tar@Cur. Finally, the addition of BP resulted in the formation of BP-PEG-Tar@Cur through a phosphorus–oxygen bond reaction and static adsorption effect. The TEM images in Figure 1A show the morphological characteristics of BP and BP-PEG-Tar@Cur. The Malvern Zetasizer Pro was used to evaluate the zeta potential, size distribution density, and stability of the nanodrug in different solvents. As shown in Figure 1B, BP exhibits an inherent potential of approximately −64.42 mV, which shifts to −25.32 mV upon combination with positively charged PEG-Tar@Cur, confirming successful synthesis. The particle size density distribution plot of the material is shown in Figure 1C. Initially, the BP nanosheets measured approximately 280 nm, but their size reduced to approximately 210 nm after modification and folding. This reduction is beneficial for enhancing circulation and metabolism of the material in vivo. Additionally, the stability of BP-PEG-Tar@Cur in PBS and DMEM was evaluated (Appendix A). The data show that BP-PEG-Tar@Cur maintains good stability in both media for over 7 days.

For deeper insight into the nanodrug synthesis, UV–vis spectroscopy, Raman spectroscopy and FT-IR spectroscopy were used for comprehensive characterization. As shown in Figure 1D, the UV–vis spectra show that Cur has a maximum absorbance peak at 425 nm. After combination with PEG-tar, the peak shifts to 382 nm, indicating the successful synthesis of PEG-Tar@Cur. Furthermore, the addition of BP to PEG-Tar@Cur results in the appearance of a new absorbance peak at 378 nm, demonstrating the successful integration of BP with PEG-Tar@Cur. The Raman spectrum of BP-PEG-Tar@Cur in Figure 1E shows characteristic peaks aligned with both BP at 361 cm^−1^ and 465 cm^−1^ and PEG-Tar@Cur at 667 cm^−1^. This alignment indicates the successful synthesis of BP-PEG-Tar@Cur. In addition, the FT-IR spectrum is shown in Figure 1F, which provides further evidence. The spectrum of PEG-Tar appears a peak at 3460 cm^−1^ corresponding to -NH, indicating the successful reaction between -COOH and -NH_2_ activation via EDC/NHS chemistry. Moreover, the strong absorbance band at 1095–1107 cm^−1^ underlines the presence of the P-O bond, confirming the integration of BP and PEG-Tar@Cur. Collectively, these spectroscopic techniques provide robust evidence for the successful synthesis and integration of BP-PEG-Tar@Cur, underscoring the precision and reliability of our novel nanodrug.

In order to standardize drug release calculations, standard curves were derived by diluting Cur and PEG-Tar@Cur at varying concentrations (Figure 1G,H). These curves served as the foundation for accurate statistical analysis of drug release kinetics. The encapsulation and drug loading efficiency of Cur within BP-PEG-Tar@Cur were determined to be 22.75% and 20.6%, respectively, based on the aforementioned curves. It is noteworthy that this loading efficiency was 4% higher than that of the curcumin-loaded nanodrug prepared by an alternative method [28]. In addition, the in vitro drug release profile of BP-PEG-Tar@Cur, depicted in Figure 1I, revealed an initial rapid release phase within the first 6 h, followed by a plateau, ultimately achieving a total release efficiency of 45%. Moreover, as evident from the data presented in the figure, the drug release amount of BP-PEG-Tar@Cur is higher than that of PEG-Tar@Cur, demonstrating the advantages of BP as a carrier. To assess the photothermal characteristics of the nanodrug, solutions with varying concentrations of BP-PEG-Tar@Cur were exposed to an 808 nm NIR laser (1.0 W/cm^2^) for 10 min, with PBS serving as the control. As demonstrated in Appendix A, the 0.5 μg/mL concentration exhibited superior photothermal effects compared to PBS, manifesting a more pronounced temperature increase. Notably, at a concentration of 20 μg/mL, BP-PEG-Tar@Cur elicited an even more dramatic temperature surge of approximately 12 °C within 10 min. Additionally, the results of the in vivo photothermal effect are presented in Appendix A. Thermal imaging maps revealed that the temperature in the control group remained stable, whereas in the two groups receiving NIR treatment, the group treated with BP-PEG-Tar@Cur and NIR exhibited a larger temperature change compared to the pure NIR group, with a temperature difference of up to 2.9 °C. Furthermore, photographic documentation of the extracted brains revealed that the control group and the pure NIR group exhibited minimal Evans blue penetration. However, the BP-PEG-Tar@Cur +NIR group displayed a noticeable blue distribution, confirming the photothermal effect of BP-PEG-Tar@Cur. These remarkable photothermal properties of BP-PEG-Tar@Cur provide a promising foundation for future in vivo BBB opening studies.

### 3.2. Biocompatibility Assessment

Prior to undertaking experimental work, it is essential to assess the biocompatibility of the synthesized materials. This evaluation provides preliminary insight into potential toxicological profiles and serves as a basis for determining the feasibility of subsequent experiments. To this end, a cell counting kit-8 (CCK8) assay was employed to quantify the viability of mouse-derived neuroblastoma cells (N2 a) cells and mouse brain microvascular endothelial cells (bEnd.3) cells upon exposure to different concentrations (0.1, 0.5, 1, 2, 4 μg/mL) of three stage-specific products for 48 h and 72 h, respectively (Figure 2A–D). The results demonstrated that the cell viability for all treated groups remained above 95%, indicative of excellent biocompatibility for the nanodrug at concentrations up to 4 μg/mL. This result determined the suitability of the nanodrug for further experimentation.

### 3.3. BP-PEG-Tar@Cur Effectively Inhibits Aβ Aggregation and Dissociates Aβ Fibrils

Aβ is widely acknowledged as a crucial element in the pathogenesis of AD within the scientific community [36]. This peptide has a proclivity to aggregate, forming profibrils which subsequently evolve into amyloid plaques, ultimately leading to neuronal damage [37]. Current therapeutic strategies targeting Aβ primarily seek to disrupt its aggregation process, thus preventing the development of harmful substances.

The inhibitory effect of the synthesized materials on Aβ aggregation was initially evaluated through a soluble Aβ experiment. Soluble Aβ monomers were incubated with various materials (Cur, PEG-Tar@Cur, BP-PEG-Tar@Cur) for 72 h. In the absence of treatment, Aβ monomers spontaneously aggregated into profibrils. Subsequently, the incubated mixtures were centrifuged and the remaining soluble monomers were quantified by BCA in order to evaluate the inhibitory capacity of each compound. As illustrated in Figure 2E, the levels of soluble Aβ in the untreated control group decreased to 32.86%, while in the Cur-, PEG-Tar@Cur-, and BP-PEG-Tar@Cur-treated groups, the levels were at 54.06%, 67.77%, and 76.51%, respectively. These findings suggest that the tested compounds exhibit inhibitory effects on Aβ aggregation, with BP-PEG-Tar@Cur demonstrating the most significant inhibitory activity.

To further validate our previous findings, we conducted a Thioflavin-T (ThT) fluorescence assay. ThT is an exogenous fluorescence probe that exhibits a high degree of affinity for Aβ fibrils. The fluorescence intensity is proportional to the beta sheet structure of the fibrils [32]. This enabled an indirect assessment of the quantity of Aβ fibrils based on the fluorescence intensity. Aliquots of the mixed solution were extracted at different time intervals for the purpose of analyzing the tendency of Aβ protofibril formation. As illustrated in Figure 2F, minimal alternations in fluorescence were discerned during the initial stages (up to 4 h), indicating a minimal level of Aβ protofibrils formation. However, 4 h later, the aggregation process accelerated rapidly and reached a plateau phase after 12 h. Notably, the PEG-Tar@Cur- and BP-PEG-Tar@Cur-treated groups reached this plateau more rapidly than both the untreated and Cur groups, suggesting at their potential for curtailing Aβ aggregation duration. Additionally, the fluorescence intensity for the BP-PEG-Tar@Cur group was reduced by 50% at 24 h in comparison to the untreated group, indicating its excellent inhibitory capacity against Aβ aggregation.

Furthermore, the capability of BP-PEG-Tar@Cur to dissociate Aβ protofibrils was assessed through Aβ fluorescence-based depolymerization experiments. After 24 h of co-incubation with Cur and BP-PEG-Tar@Cur, the formation of Aβ protofibrils was observed under a fluorescence microscope. Images were randomly captured under the microscope and photographed, as presented in Figure 2G. The formation of protofibrils in the untreated group was observed to occur in densely populated large patches. In contrast, both the Cur and BP-PEG-Tar@Cur treatments were found to considerably reduce the density of these patches. Remarkably, the BP-PEG-Tar@Cur group displayed only minor patches, suggesting its efficacy in dissociating Aβ protofibrils. These findings provide compelling evidence that BP-PEG-Tar@Cur not only effectively suppresses aggregation of Aβ, but also possesses a remarkable ability in dissociating Aβ protofibrils.

### 3.4. BP-PEG-Tar@Cur Enhances the Ability of Aβ Targeting and Cellular Uptake

Cur is renowned for its exceptional biological activity, yet its applications have been hindered by low solubility and suboptimal bioavailability. To address these limitations, we modified Cur with PEG to enhance its biocompatibility and solubility. In addition, to enlarge the target ability of the nanodrug, 4-(Dimethylamino) cinnamic acid was introduced as a mediator for Aβ targeting. Cinnamic acid, an aromatic carboxylic acid, is recognized for its multifarious biological activities such as anti-inflammatory, antioxidant, and anticancer properties [23]. Previous research combining cinnamic acid and curcumin has demonstrated promising results [21]. Building upon this existing literature, we found that ThT fluorescence binds to fibers due to its co-folding with the beta sheet through its dimethylamine group [24]. Inspired by this, we adopted a 4-(Dimethylamino) cinnamic acid, structurally akin to ThT, in order to achieve targeting. To evaluate the targeting efficacy, the samples of Cur, BP-PEG@Cur, PEG-Tar@Cur, and BP-PEG-Tar@Cur were incubated with Aβ fibrils and visualized using confocal fluorescence microscopy. The resulting images (Figure 3A) indicate the presence of Aβ in red and Cur in green. Remarkably, after the addition of PEG-Tar@Cur and BP-PEG-Tar@Cur, an extensive overlapping area of green and red is discernible–significantly more than with Cur and BP-PEG@Cur group. This experimental result validates our hypothesis that the Tar has the ability to enhance Aβ-targeting ability. Furthermore, statistical analysis revealed that the fluorescence intensity in the compound group with Tar was fourfold that of pure Cur group, indicating the superior cellular uptake proficiency of BP-PEG-Tar@Cur.

### 3.5. BP-PEG-Tar@Cur Effectively Removes Mitochondrial ROS In Vitro

To investigate the antioxidant properties of the synthesized nanodrug, a co-incubation study was conducted, whereby both Cur and BP-PEG-Tar@Cur were incubated with Aβ fibrils. Subsequently, the levels of mitochondrial ROS were quantified using the Mito-SOX mitochondrial red fluorescence probe (Figure 4A). The treatment of cells with Aβ fibrils resulted in the generation of extensive red fluorescence, indicating a pronounced ROS generation by the induction of Aβ fibrils. As has been demonstrated in previous studies, the group treated with Cur exhibited a reduction in ROS levels. However, the most noteworthy observation was that cells treated with BP-PEG-Tar@Cur exhibited a marked decrease in red fluorescence, approximately one-third lower than those of the untreated group. These results suggest that BP-PEG-Tar@Cur effectively curtails mitochondrial ROS levels induced by Aβ aggregation. The evidence also spotlighted the synergistic benefits derived from cinnamic acid, which elevated the efficacy of BP-PEG-Tar@Cur beyond that of the Cur alone.

### 3.6. BP-PEG-Tar@Cur Effectively Opens BBB Under NIR Irradiation and Enhances Drug Permeability

The BBB has historically been regarded as a significant obstacle in AD research, frequently impeding the effective delivery of therapeutic drugs to the brain and consequently limiting their therapeutic efficacy. In this study, we employed the photothermal properties of BP nanosheets under NIR irradiation to transiently open the BBB, thereby creating a temporary drug window that facilitates drug permeability. To validate the efficacy of BBB penetration in vitro, an in vitro BBB model was constructed using a transwell device (Figure 5B). In the upper chamber of the device, bend.3 cells were seeded in order to simulate human brain vascular endothelial cells. Once the optimal cell density had been reached, the trans-epithelial electrical resistance (TEER) was measured using a cell resistance meter. Equal concentrations of Cur and BP-PEG-Tar@Cur were added to the upper chamber once the TEER reached 200 Ω·cm^2^. The NIR group was subjected to 10 min of NIR irradiation (808 nm, 1 W/cm^2^) following the administration of the drug. Subsequently to a 2 h incubation period, samples from the lower chamber were collected and analyzed, and the penetration rate (illustrated in Figure 5A) was calculated. The results demonstrated that the penetration rates of Cur and BP-PEG-Tar@Cur+NIR were 8.13% and 36.84%, respectively. These findings highlight the efficacy of BP-PEG-Tar@Cur in facilitating BBB opening under NIR irradiation. After giving NIR irradiation and drug treatment, the resistance value of each group was measured and recorded (Appendix A). The change in the resistance was negligible, with a slight upward trend. It is hypothesized that this may be a self-regulating feedback mechanism that is activated in order to enhance protective capabilities following any potential damage. Therefore, it is inferred that the short-term NIR irradiation may not compromise the integrity of the BBB. To further verify whether NIR irradiation caused cell damage, the calcein acetoxymethyl ester/propidium iodide (calcein-AM/PI) assay was utilized. The cells extracted from the transwell model experiment were digested and resuspended, and a calcein-AM/PI working solution was subsequently added. In healthy cells, calcein-AM is converted to the fluorescent molecule calcein, which indicates cell viability. In contrast, PI stains dead or damaged cells, emitting red fluorescence. The fluorescence images (Appendix A) revealed no significant signs of cell damage in the NIR-treated group, indicating that NIR irradiation did not cause appreciable cellular harm.

### 3.7. BP-PEG-Tar@Cur+NIR Alleviates the Burden of Aβ and Associated Proteins in AD Mice

To investigate the in vivo therapeutic efficacy of the synthesized nanodrug, we selected APP/PS1 mice as an AD model. After 3 weeks of drug treatment, the hippocampal tissue was extracted and subjected to quantitative analysis via the Western blot (WB) assay. As is widely known, APP is a precursor protein of Aβ. BACE1 and PS1 are crucial enzymes in the generation of Aβ through the APP cleavage process [7]. In order to study the impact of BP-PEG-Tar@Cur on the Aβ production pathway, APP, Aβ, BACE1, and PS1 sites were selected for investigation. The protein bands are shown in Figure 5C, with their corresponding statistical analysis results in Figure 5D–G. As illustrated in the figure, the protein expression levels in the APP/PS1 mice were markedly elevated in comparison to the wild-type group, confirming the authenticity of the AD model. Notably, the mice treated with Cur and BP-PEG-Tar@Cur + NIR exhibited reduced Aβ levels, with the latter displaying a more pronounced decline. This phenomenon was also observed in the other three proteins, indicating that both Cur and BP-PEG-Tar@Cur + NIR can reduce the deposition of Aβ and its related proteins accumulated in the hippocampus of AD model mice. Furthermore, the outcomes achieved with BP-PEG-Tar@Cur + NIR were superior.

### 3.8. BP-PEG-Tar@Cur+NIR Mitigates the Inflammatory Response Induced by Microglia and Astrocytes

We conducted a more thorough investigation into the potential of BP-PEG-Tar@Cur + NIR in mitigating inflammatory responses within the mouse brain. Given that microglia and astrocytes are the primary immune cells implicated in AD progression, recent studies suggest that reducing their activation may alleviate AD symptoms [29]. Therefore, in our investigation, we chose Iba-1 as a microglia marker and GFAP as an astrocyte marker to quantitatively analyze the inflammatory response. Following the conclusion of the treatment period, the mouse brain tissue was subjected to immunostaining with GFAP and Iba-1 antibodies, with observations conducted via a fluorescence microscopy. As shown in Figure 6A,C, the AD model mice exhibited a pronounced red fluorescence, which underscored the significant activation of microglia and astrocytes. However, the brains of mice treated with either Cur or BP-PEG-Tar@Cur+NIR resulted in diminished red fluorescence, indicating a significant reduction in the inflammatory response caused by the activation of these immune cells. Although BP-PEG-Tar@Cur+NIR demonstrated superior efficacy in clearing the inflammatory response, the numerical difference between it and Cur was not as substantial as anticipated. This may be attributed to the potential combined effect of curcumin and cinnamic acid in BP-PEG-Tar@Cur in reducing inflammation. However, as illustrated by the statistical data in Figure 6B,D, the number of positive cells in the BP-PEG-Tar@Cur group was nearly equivalent to that of the wild-type group. This suggests that a potential bottleneck effect may have contributed to the apparent lack of significant anti-inflammatory effects of BP-PEG-Tar@Cur+NIR.

### 3.9. BP-PEG-Tar@Cur+NIR Effectively Improves Memory and Behavioral Impairment in AD Mice

A prominent pathological characteristic of AD is cognitive dysfunction, typified by deteriorating memory and learning capabilities. The water maze test is the most commonly used method to evaluate spatial memory and learning ability in mice [38]. Therefore, following the completion of the drug treatment, the mice were subjected to a week-long training session on the water Morris maze, with their movements being analyzed using a video analysis software (2860). The distinct swimming trajectories depicted in Figure 7A illustrate notable differences among the groups. It is obvious that the wild-type mice displayed more purposeful routes towards the platform (crossed the platform more than three times), while the AD mice meandered aimlessly (never crossed the platform). Both the Cur and BP-PEG-Tar@Cur + NIR groups demonstrated improvement, with the latter exhibiting superior memory retention. The escape latency, which is defined as the time taken by mice to locate the platform upon entering the water, serves as a crucial indicator of memory performance. A shorter escape latency indicates better memory function. All treatment groups exhibited a reduction in escape latency over the 5-day training period (Figure 7B). Especially by the end of training, the BP-PEG-Tar@Cur + NIR group exhibited pronounced improvements in memory and learning ability. The superiority of the wild-type groups was evident, with a shorter escape latency than the other groups. Notably, the BP-PEG-Tar@Cur + NIR groups had a similar escape latency to the AD group at the outset of training, but demonstrated a marked reduction by the conclusion of training, indicating a certain degree of improvement in learning and memory ability. There was no significant difference in the swimming speed among the four treatment groups, suggesting that the observed variations in results were not attributed to exercise deficits (Figure 7D). The mean escape latency, which is the average value of escape latency, reflects the accuracy of the data (Figure 7C). The frequency of crossings over the platform serves as an additional indicator of memory performance in mice (Figure 7E). In addition, a longer duration spent within the platform’s quadrant suggested superior spatial memory (Figure 7F). The collective data demonstrated a substantial improvement in spatial learning and memory performance in mice treated with BP-PEG-Tar@Cur+NIR, suggesting the promising effect of the synthesized nanodrug in ameliorating behavioral deficits in AD mice.

### 3.10. In Vivo Safety Evaluation

To ascertain the biocompatibility and potential toxicity of our synthesized nanodrug, we conducted an in vivo analysis of its impact on visceral organs using hematoxylin and eosin (HE) staining. During the post-experimental period, organs—brains, hearts, livers, spleens, lungs and kidneys—were harvested from the mice that had been assigned to various treatment groups, and processed for HE staining. As illustrated in Figure 7G, the histological analysis reveals no discernible indications of tissue damage in the treated organs after treatment in comparison to the control group. This observation strongly indicates that neither Cur nor BP-PEG-Tar@Cur induced toxic reactions after intravenous injection. Thus, the synthesized nanodrug has been demonstrated to exhibit low toxicity and possess robust biocompatibility, paving the way for further preclinical and clinical evaluations.

## 4. Conclusions

In conclusion, a nanodrug that is specifically designed to target Aβ was reported to provide synergy treatment for AD. In addition to inhibiting the self-assembly of Aβ, this nanodrug has also been observed to dissociate Aβ plaques. In particular, compared with pure Cur, BP-PEG-Tar@Cur possessed superior cellular absorbance and Aβ-targeting capability. Furthermore, the antioxidant properties of BP-PEG-Tar@Cur effectively scavenged Aβ-induced ROS. The nanodrug exhibited effective penetration of the BBB under the NIR irradiation, and the behavioral test proved that spatial memory and learning deficits in AD mice were improved following treatment. The present research surveys the biological application of a novel nanodrug and provides new insights for developing multifunctional photothermal agents as a potential therapy for AD.

## Figures and Tables

**Figure 1 pharmaceutics-17-00177-f001:**
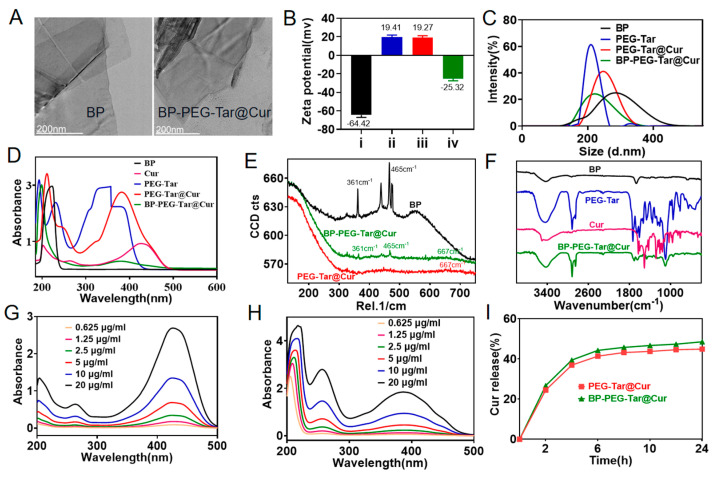
Characterization of the synthesized nanodrug. (**A**) The TEM images of BP (scale bar: 200 nm) and BP-PEG-Tar@Cur (scale bar: 200 nm). (**B**) The zeta potential of various materials (i: BP; ii: PEG-Tar; iii: PEG-Tar@Cur; iv: BP-PEG-Tar@Cur). (**C**) The particle size density distribution of various particles. (**D**) The UV absorbance spectrograms of different materials. (**E**) The Raman spectroscopies of different materials. (**F**) The Fourier transform infrared spectroscopies of various particles. The standard curves of Cur (**G**) and PEG-Tar@Cur (**H**). (**I**) The in vitro drug release curve of BP-PEG-Tar@Cur and PEG-Tar@Cur.

**Figure 2 pharmaceutics-17-00177-f002:**
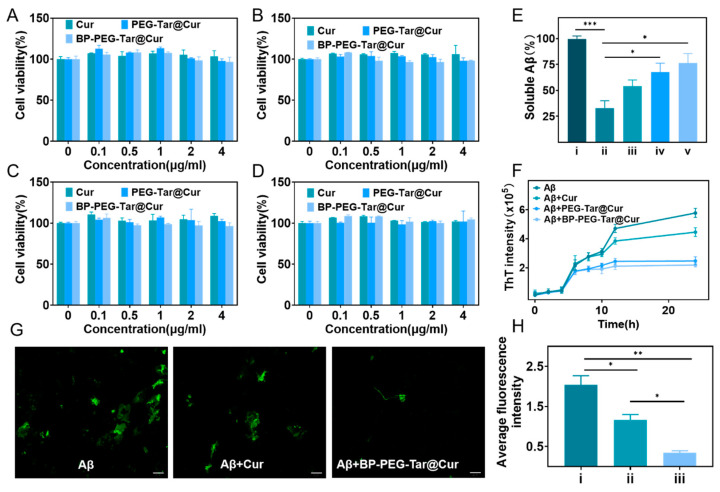
The biocompatibility assay and the effect of nanodrug inhibiting Aβ aggregation and dissociating Aβ fibrils in vitro. Results of cell viabilities by CCK8 assays: (**A**) N2 a incubated 48 h; (**B**) N2 a incubated 72 h; (**C**) bEnd.3 incubated 48 h; (**D**) bEnd.3 incubated 72 h. (**E**) The data of soluble Aβ experiment (i: Aβ monomers; ii: Aβ fibrils; iii: Aβ monomers + Cur; iv: Aβ monomers + PEG-Tar@Cur; v: Aβ monomers + BP-PEG-Tar@Cur). (**F**) The Thioflavin-T (ThT) fluorescence assay of Aβ monomers treated with Cur, PEG-Tar@Cur and BP-PEG-Tar@Cur. (**G**) The Aβ depolymerization on fluorescence assay (Scale bar: 50 μm). (**H**) Quantified fluorescence intensities of (i: Aβ fibrils; ii: Aβ fibrils + Cur; iii: Aβ fibrils + BP-PEG-Tar@Cur). Data as mean ± SEM, *n* = 3. (*p* < 0.05 *, *p* < 0.01 **, *p* < 0.001 ***).

**Figure 3 pharmaceutics-17-00177-f003:**
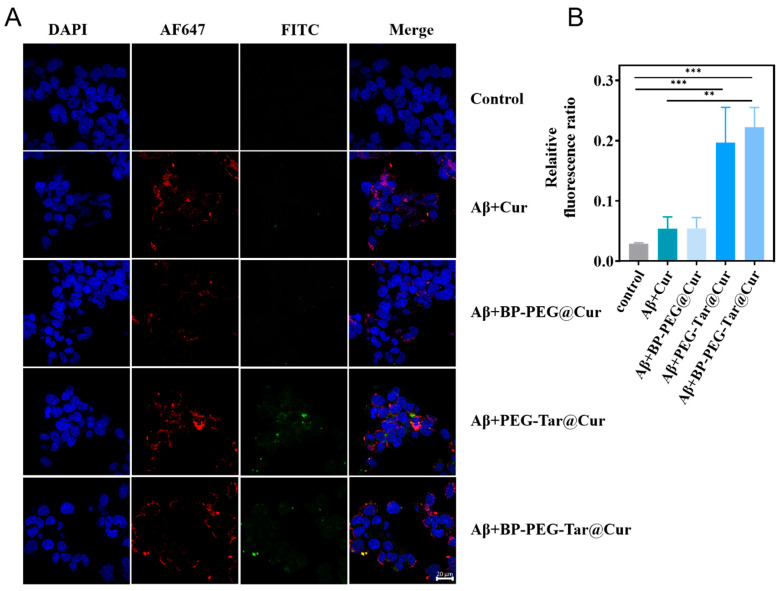
BP-PEG-Tar@Cur enhances Aβ-targeting ability. (**A**) Representative fluorescence images of the Aβ-targeting ability assessment of Cur, BP-PEG@Cur, PEG-Tar@Cur, and BP-PEG-Tar@Cur under confocal fluorescence. (Aβ, AF647 in red channel; Cur in green channel) (Scale bar: 20 μm). (**B**) Quantitative relative fluorescence ratio of (**A**). Data as mean ± SEM, *n* = 3. (*p* < 0.01 **, *p* < 0.001 ***).

**Figure 4 pharmaceutics-17-00177-f004:**
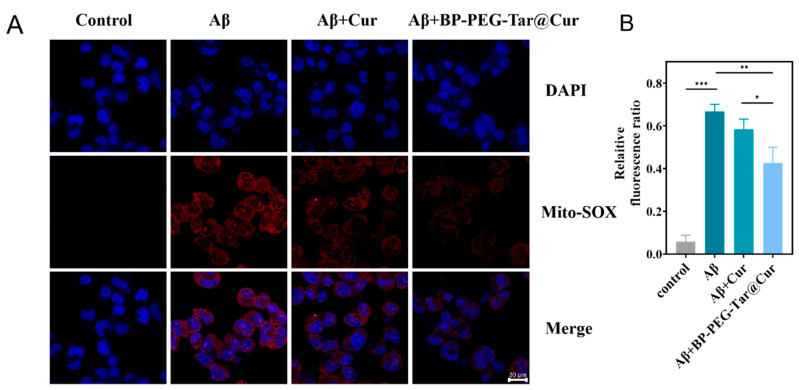
BP-PEG-Tar@Cur enhances mitochondrial ROS clearance ability. (**A**) Representative fluorescence images of the mitochondrial ROS clearance. (**B**) Quantitative relative fluorescence ratio of (**A**). Data as mean ± SEM, *n* = 3. (*p* < 0.05 *, *p* < 0.01 **, *p* < 0.001 ***).

**Figure 5 pharmaceutics-17-00177-f005:**
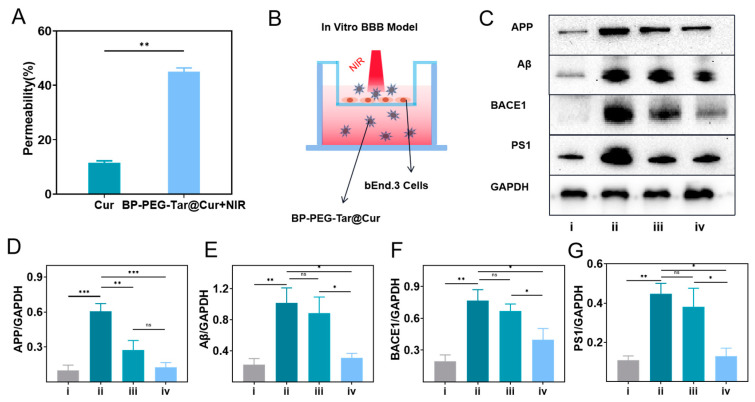
In vitro BBB transwell model and BP-PEG-Tar@Cur+NIR alleviate the burden of Aβ-related proteins in the hippocampal tissue of the mice. (**A**) The permeability of Cur and BP-PEG-Tar@Cur+NIR. (**B**) The schematic of in vitro BBB model. (**C**) Protein bands of APP, Aβ, BACE1, PS1, and GAPDH in each group, respectively. GAPDH served as the internal control (i: control; ii: AD; iii: AD + Cur; iv: AD+BP-PEG-Tar@Cur + NIR). Quantitative analysis of protein blot grayscale value in APP (**D**), Aβ (**E**), BACE1 (**F**), PS1 (**G**). Data as mean ± SEM, *n* = 3. (*p* < 0.05 *, *p* < 0.01 **, *p* < 0.001 ***, ns: not significant).

**Figure 6 pharmaceutics-17-00177-f006:**
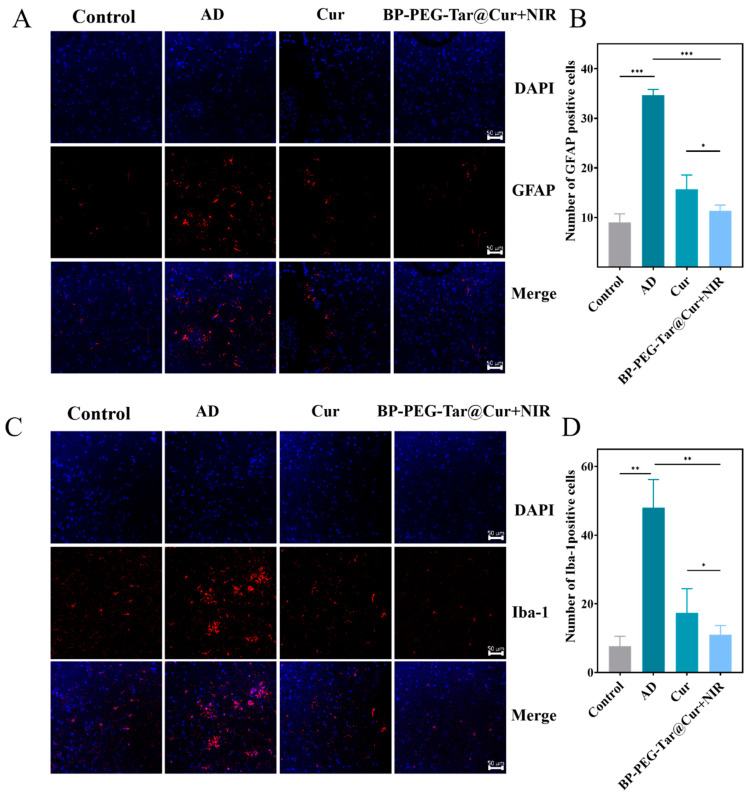
BP-PEG-Tar@Cur+NIR mitigates the inflammatory response induced by microglia and astrocytes. Immunofluorescence for astrocyte activation marker GFAP (**A**) and microglial activation marker Iba-1 (**C**) in the mouse brain slices. (**B**,**D**) Quantitative analysis of (**A**,C), respectively (scale bar: 50 μm). Data as mean ± SEM, *n* = 3. (*p* < 0.05 *, *p* < 0.01 **, *p* < 0.001 ***).

**Figure 7 pharmaceutics-17-00177-f007:**
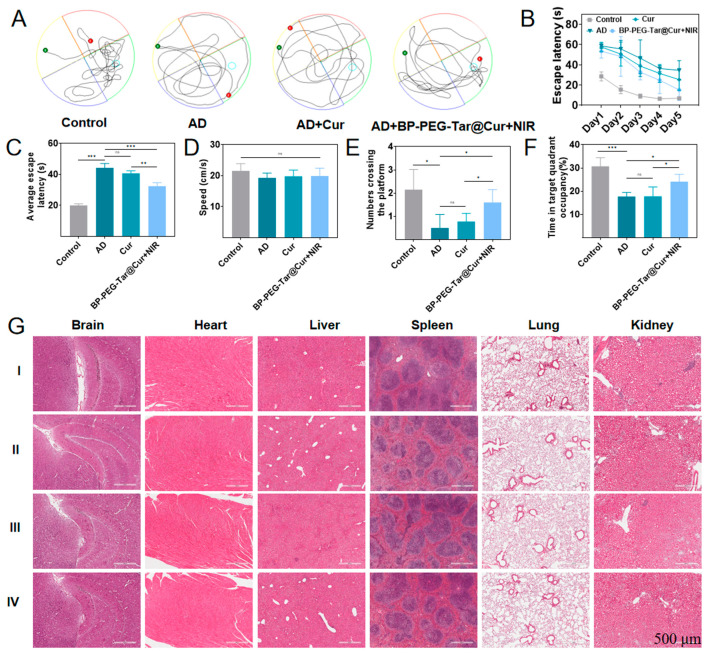
BP-PEG-Tar@Cur+NIR treatment improved cognitive function in APP/PS1 mice assessed by Morris water maze and the results of hematoxylin and eosin (HE) staining after training. (**A**) The swimming trajectories of mice treated in different groups. (The green dot represents the entry point of the mice into the water, the red dot represents the position of the mice at the end of the experiment, and the light blue circle represents the location of the virtual platform.) (**B**) The escape latency of the mice in different treatment groups on each day for five days. (**C**) The average escape latency for five days. (**D**) The swimming speeds of each group. (**E**) The numbers crossing the platform with the platform removed. (**F**) The time spent in the target quadrant. (**G**) The images of HE staining after treatment. I: control; II: AD; III: Cur; IV: BP-PEG-Tar@Cur + NIR (scale bar: 500 μm). Data as mean ± SEM, *n* = 3. (*p* < 0.05 *, *p* < 0.01 **, *p* < 0.001 ***, ns: not significant).

## Data Availability

The data generated during and/or analyzed during the current study are available from the corresponding author on reasonable request.

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
