# Peer review of "Advanced Nanopharmaceutical Intervention for the Reduction of Inflammatory Responses and the Enhancement of Behavioral Outcomes in APP/PS1 Transgenic Mouse Models"

_pharmaceutics, 2025, doi:10.3390/pharmaceutics17020177_

Round 1

Reviewer 1 Report

Comments and Suggestions for Authors

The authors have presented the results of a thorough investigation of a new drug against Alzheimer’s disease. To my mind, the methods of investigations are correct, and the obtained results are interesting. The most important innovation is the use of black phosphorus nanosheets as a platform to combine all components and simultaneously weaken the blood brain barrier via laser induced heating. I recommend to publish the manuscript after minor revisions. My comments are given below.

line

text

Comments

155

After drying, the sample was photographed under the condition of an accelerating voltage of 200 kV.

TEM does not produce photographs.

‘photographed’ should be replaced with ‘imaged’

158

ATR

acronym should be explained ‘attenuated total reflectance’

161, 403

4cm-1

465 cm-1, 667 cm-1.

4 cm^{-1} superscript

199

424

To investigate the temperature change of BP-PEG-Tar@Cur under NIR irradiation in vitro, different concentrations of BP-PEG-Tar@Cur (0, 0.5, 2, 5, 10, 20 μg/ml) in PBS were irradiated by NIR (1 W/cm2) for 10 min.

NIR is a wide spectral range. Please, give the characteristics of the light source used.

Information from the line 424 should be included here

208

424

NIR laser probe

What was the wavelength of the laser? Information from the line 424 should be included here

278

irradiation was administered 10 min after the addition

How long was irradiation in this case?

280

The status of cells after NIR irradiation

 Information from the line 424 should be included here

371

Interventionary studies involving animals or humans, and other studies that require 371 ethical approval, must list the authority that provided approval and the corresponding ethical approval code.

What for this statement is included in the text of the article?

375

3.1. Subsection

What is the title of this subsection?

399

Furthermore, the addition of BP to PEG-Tar@Cur results in a red shift from 200 nm to 223 nm,

At this stage it is not clear which substance has had a band at 200 nm before BP was added

647

Fig.1 C The particle size density distribution of various particles.

What do the negative values of ‘intensity(%)’ in this graph mean?

650

The 649 standard curves of Cur (G) and PEG-Tar@Cur (H).

Please, give an exact meaning of ‘Absorption’ in this graph. Is it dimensionless “Absorbance” as it is defined in  https://en.wikipedia.org/wiki/Absorbance

Otherwise, please, give the dimensions.  

Reviewer 2 Report

Comments and Suggestions for Authors

Dear Dr. Li,

I have an honor to review your manuscript titled “Advanced Nanopharmaceutical Intervention for the Reduction of Inflammatory Responses and the Enhancement of Behavioral Outcomes in APP/PS1 Transgenic Mouse Models”

Your work is devoted to the actual topic of searching and development of new effective methods and drugs for the treatment of Alzheimer's disease.

The paper is written in good scientific language and is well structured. The Materials and Methods section is thoroughly and very detailed, and the steps of the nanoparticle preparation process are well described. The course of the experiments is also described in an extremely meticulous manner.

 While studying the work, I had a number of comments, the response and correction of which, I believe, can improve the quality and perception of the text of the work.

 The remarks concern both issues of text design and some terminological issues.

Your work and the materials and experiments used are rather difficult to perceive for an untrained reader who is not deeply versed in these issues. In this connection, it is of fundamental importance to bring all abbreviations into conformity - unfortunately, many of them are not deciphered, as, for example, in lines 118-119, 178,256, 268, 564, 715. I think it is extremely important for a better perception of the text to give the decoding of all abbreviations, even if you consider them widely applicable (e.g. PBS), at the end of the paper, where the decoding of some abbreviations is given.

There are a number of typos and inaccuracies (ambiguities) in the paper.

In line 48 you talk about abnormal radicals. What kind of radicals are you referring to? Probably oxygen free radicals, nitrogen free radicals? It seems to me that in describing the pathogenesis of Alzheimer's disease it is more correct to talk about specific substances, and probably you mean reactive forms of oxygen and nitrogen  (ROS and RNS) leading to the development of oxidative stress (they are not only radical in nature). Please make it clear.

 Line 178 is a typo (‘Subsequetly’)

 The sentence in lines 371-373 looks like an excerpt from the general rules or an undeleted draft.

 Line 451 looks like the singular and plural are mixed up (inhibit(s), dissociate (s)).

The fundamental issue in my opinion is the terminology in line 515-516. 

What is antioxidant stress? What does the term capacity refer to? To the antioxidant properties of curcumin-based nanodrugs? If so, it is crucial to understand whether we are talking about them in the context of kinetics or thermodynamics (there is a very large confusion in the scientific literature in these terms regarding the antioxidant properties of various substances). Do you mean antioxidant properties of nanodrug against oxidative stress/oxidative damage? If you mean capacity, in what units do you measure it?

 On line 544, you give a value of 8.129%. The third decimal place looks like an unmeasurable value. Is this really appropriate, maybe it is possible to round to two decimal places?

The conclusions of your work generally correspond to the goals and objectives of the experiments conducted. But there is a terminological inaccuracy in lines 707-708. You use the term activity. This is a thermodynamic term. Before that, you used the term capacity in lines 515-516. This creates some terminological confusion. Apart from all of the above, “activity” cannot scavenge ROS. Rather, we can say that the antioxidant properties of the nanodrug allow it to effectively scavenge reactive oxygen species. In order to measure the antioxidant activity of the drug, a separate experiment is needed.

 I believe that once these remarks have been addressed, your work can be recommended for publication.

Reviewer 3 Report

Comments and Suggestions for Authors

The article is well presented, with a considerable number of figures to support the results presented. However, for a better final version of the article, some minor points should be addressed.

1) The abstract should be written in a continuous form without naming sections such as background, introduction, methodology, etc. Please check the format given in the author guidelines section.

2) In the Introduction, it is mentioned that Figure 1B illustrates the partial therapeutic effects of this nanodrug. However, Figure 1B shows the zeta potential of several materials (i: BP; ii: PEG-Tar; iii: PEG-Tar@Cur; iv: BP-PEG-Tar@Cur). Does the figure mentioned in the introduction exist?

3) Into the characterization section (page 4 between lines 61-62) there are typographical errors: the wavenumber units of the FT-IR section should be superscripted.

4) Is this heading correctly named? “3.1. Sub-section”

5) Report other more representative TEM images where the morphological characteristics of BP and BP-PEG-Tar@Cur (nanosheets) are clearly observed.  The scale in Figure 1 is not observable and are not the best images showing the nano-formulation.

6) Section 3.2. refers to figures, tables and schemes. No schemes and no tables are reported, change heading to 3.2. Figures only.

7) It would also be good to include the graphical summary in this section.

8) In my opinion, figures 1G and 1H are not necessary or, in your case, send them as supplementary figures.
